

# Spey: Smooth inference for reinterpretation studies

 Jack Y. Araz⋆

Institute for Particle Physics Phenomenology, Durham University, Durham, UK

⋆ jack.araz@durham.ac.uk

## Abstract

Statistical models serve as the cornerstone for hypothesis testing in empirical studies. This paper introduces a new cross-platform Python-based package designed to utilize different likelihood prescriptions via a flexible plug-in system. This framework empowers users to propose, examine, and publish new likelihood prescriptions without developing software infrastructure, ultimately unifying and generalising different ways of constructing likelihoods and employing them for hypothesis testing within a unified platform. We propose a new simplified likelihood prescription, surpassing previous approximation accuracies by incorporating asymmetric uncertainties. Moreover, our package facilitates the integration of various likelihood combination routines, thereby broadening the scope of independent studies through a meta-analysis. By remaining agnostic to the source of the likelihood prescription and the signal hypothesis generator, our platform allows for the seamless implementation of packages with different likelihood prescriptions, fostering compatibility and interoperability.

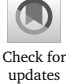

# 1 Introduction

Unraveling new physics beyond the Standard Model has been a pursuit driving scientists to expand the boundaries of experimental analyses. The forefront of this exploration is the Large Hadron Collider (LHC), where valuable analyses are conducted, specifically targeting phase spaces to capture the signature of simplified models. These models, such as variations of the minimal supersymmetric extension of the Standard Model (MSSM), often include fixed masses and decay branching ratios, albeit unable to fully encapsulate the effects of a complete theory. While these searches yield crucial insights, the ever-expanding theoretical landscape presents scenarios beyond the reach of simplified models.

In response to this challenge, various methods have emerged to reinterpret and broaden the application of experimental analyses. These strategies encompass the reinterpretation of analyses using Monte-Carlo (MC) and detector simulations [1–10] as well as simplified approaches utilising efficiency maps [11].

Analyzing the vast amounts of data from the LHC involves statistical modeling to distill meaningful information from experimental results, enabling inferences about new physics. While recent strides have been made in publishing complete statistical models, most analyses offer limited information, prompting the emergence of approximated approaches to capture the likelihood distribution of the original statistical models [12–16]. Particularly in cases with limited correlation information, simplistic Gaussian toy model-based approaches have been utilized (see ref. [17] for an example). Hence, inconsistencies arise in the publication of statistical models or data to construct them, leading to inaccurate inference of the new theories. The recent efforts by the HS3 collaboration[1] show promising advancements in standardization of likelihoods. However, numerous applications continue to rely on statistical models that fall outside the standardized prescriptions of the HS3 proposal, posing a significant challenge in their utilization.

This study introduces a new cross-platform Python-based package, SPEY.[2] This package harnesses a versatile plug-in system supporting standardized likelihood prescriptions for statistical hypothesis testing; computing exclusion limits, assessing discovery significance, and conducting parameter fitting. It accommodates various likelihood prescriptions, referred to as "backend" implementations, seamlessly integrating existing inference packages or facilitating the development of new ones. This design ensures agnosticism towards specific implementations, providing a flexible framework for likelihood inference, even for yet-to-be-proposed prescriptions.

Leveraging this flexibility, SPEY offers default approximated likelihood approaches specifically tailored for counting experiments such as those conducted at the LHC, i.e. simplified likelihood framework [13, 14]. These approaches encompass both correlated and uncorrelated Poisson-based likelihood prescriptions. However, these approaches tend to over-exclude certain parameter spaces. To mitigate this, we propose an alternative likelihood approximation incorporating asymmetric background uncertainties, yielding a more precise approximation of the original likelihood distribution.

While precise approximations improve exclusion limits, utilizing full statistical models whenever possible is preferable. To address this, we introduce a backend plug-in for the widely acclaimed `pyhf` package [18, 19], integrating its functionality within SPEY for employing full statistical models while retaining backend agnosticism.

One crucial aspect in statistical modeling is likelihood combination, facilitated by SPEY's backend agnostic infrastructure, enabling various likelihood combination techniques across any backend implemented within SPEY. We showcase this capability by extending methodolo-

---

[1]Details can be found in this GitHub repository.

[2]The name of the package is inspired by the Spey River in the Scottish Highlands.

gies from ref. [20] to enhance exclusion limits through the combination of full and approximated statistical models from different experiments, demonstrating the framework's strength in maximizing the potential of the experimental results.

This study is structured as follows: Section 2 summarizes inference through likelihoods and illustrates SPEY's application using a simple example. Section 2.1 delves into the implementation of default likelihood prescriptions for correlated histograms. Section 2.2 demonstrates the utilization of full statistical models through SPEY. Likelihood combination techniques are explored in Section 3. Finally, an example beyond LHC experiments is presented in Section 4.

## 2 Statistical models

This section briefly introduces statistical modelling and its applications within SPEY. A statistical model is based on the occurrence probability of a random variable $n \in \{n_i\}$; $P(n)$.[3] This random variable $n$ can be, for instance, observations at the LHC, which includes the randomness of the detector effects such as resolution, reconstruction efficiency etc. In continuous cases, the probability assigned to a specific value can be expressed as integration over the probability density function (PDF) given by

$$P(S) = \int_S f(x_i, \cdots) d^n x . \tag{1}$$

Here, $S$ represents the integration hyper-surface over variables $x_i \in \mathbf{X}$ and $f(x_i, \cdots)$ represents the PDF. The likelihood function is defined as the overall PDF, evaluated from the observations $x_i$ with respect to some parameters $\theta_i$; $\mathcal{L} = f(x_i, \cdots | \theta_i)$. A statistical model proposes a certain structure for the PDF, capturing the distribution of the variable $\mathbf{X}$ and its uncertainties. A generic composite likelihood can be described as [22];

$$\mathcal{L}(\mu, \theta) = \underbrace{\prod_{i \in \text{bins}} \mathcal{M}(n^i | \lambda^i(\mu, \theta | n_s, n_b))}_{\text{main}} \cdot \underbrace{\prod_{j \in \text{nui}} \mathcal{C}(\theta_j)}_{\text{constraint}} . \tag{2}$$

In this equation, $\lambda_i$ is a function of signal $(n_s)$, and background $(n_b)$ yields,[4] $\mu$ represents the parameter of interest (POI) or signal strength, $\theta$ refers to the nuisance parameters, and $\mathcal{C}$ represents the constraint terms associated with uncertainties.[5] The first term, describing the main model, accounts for the bins of the histogram (represented by $i$), while the constraint term multiplies the constraint for each nuisance parameter. In the case of a counting experiment like the LHC, the Poisson distribution is commonly used to compare observations with the modelled expectations. In a simplified scenario, uncertainties can be captured using a Gaussian constraint term. Therefore, a multi-bin histogram likelihood for a background yield $n_b^i \pm \sigma_b^i$ can be modelled as:

$$\mathcal{L}(\mu, \theta) = \prod_{i \in \text{bins}} \text{Poiss}\left(n^i | \mu n_s^i + n_b^i + \theta^i \sigma_b^i\right) \cdot \prod_{j \in \text{nui}} \mathcal{N}\left(\theta^j | 0, 1\right) . \tag{3}$$

Here, we have used a unit Gaussian as the constraint term since the nuisance parameters are standardized. This simplifies the optimization process when computing the profiled likelihood. The choice of Gaussian originates from the fact that many calibration observables, whose PDF

---

[3]A review can be found in ref. [21].

[4]Notice that $\lambda$ does not need to be only a function of signal and background yields, as a matter of fact it can be a function of any theory parameter, see section 4 for an example.

[5]For simplicity, we have excluded the additional product rule over channels in the main term.

central limits to a Gaussian, are parametrised at zero nominal rates with one sigma deviation. The specific form of the $\mathcal{L}(x_i)$ is generally unknown and depends on the available information from the experimental analyses. Therefore, it may be necessary to construct an approximate PDF description in order to compare a theory hypothesis with experimental results.

Experimental analyses, particularly at the LHC, typically encompass the exploration of simplified signal hypotheses. In order to broaden the scope of such analyses, several reinterpretation techniques have been developed, ranging from intricate detector simulations to straightforward efficiency maps. These techniques enable the examination of new theoretical hypotheses in light of existing experimental results. Reinterpretation platforms are specifically designed to generate signal yields based on the chosen histograms defined by the experimental analysis, which are then subsequently incorporated into a likelihood distribution for setting exclusion limits on the given signal hypothesis.

SPEY serves as a convenient cross-platform Python-based interface that consolidates various PDF implementations in a unified framework for reinterpretation studies. It comes equipped with various differentiable PDF prescriptions while maintaining a simplified likelihood methodology as the default approach. This means that the default PDFs included in SPEY are designed to approximate the original likelihood based on the available information regarding yields and uncertainties. It is important to note that there are multiple ways to approximate a likelihood distribution, some better than others. To address this, the API is constructed with an expandable plug-in structure at its core, allowing users to implement and publish their own PDF prescriptions independently. Once SPEY is installed, it automatically detects new plug-ins through the backend detection system and incorporates them into the inference process without requiring modifications to the main code structure. For detailed instructions on building a custom likelihood model and utilizing it within SPEY, we refer the reader to Appendix A. Additionally, guidance on appropriately citing these independent plug-ins is provided in Appendix B.

SPEY has been integrated within the Python package index and can be downloaded and imported via

```
1  !pip install spey
2  import spey
```

commands.[6] Afterwards, all available PDFs can be printed using `spey.AvailableBackends()` function. This function will list all the default backends as well as third-party plug-ins, if applicable. A list of backend constructors currently available through SPEY has been listed in Table 1.

To represent a simple statistical model described by equation (3) for a counting experiment with two bins, one can utilize the `"default_pdf.uncorrelated_background"` backend in SPEY. The PDF backend can be accessed using the `get_backend` function, as demonstrated below:

```
1  pdf_wrapper = spey.get_backend("default_pdf.uncorrelated_background
       ")
```

The variable `pdf_wrapper` is a wrapper function that ensures proper integration of the PDF prescription into SPEY by verifying various versioning and inheritance constraints.[7]

To exemplify the process, let's construct a two-bin uncorrelated statistical model, as defined by equation (3), using the following observed yields: 36 and 33. The expected background yields are given as $50 \pm 12$ and $48 \pm 16$, while the signal model yields 12 and 15. The code snippet below showcases this implementation:

```
1  statistical_model = pdf_wrapper(
```

---

[6]An online documentation can be found in this link.

[7]Refer to Table 1 for a list of accessors. The documentation of the `pdf_wrapper` function also specifies the backend it wraps around.

Table 1: List of available PDF accessors which summoned through `spey.get_backend` (`"<accessor>"`) function. The usage of each accessor has been shown in the online documentation of the package.

| Accessor | Explanation |
| --- | --- |
| `"default_pdf.uncorrelated_background"`, Arguments: signal yields, background yields, data, background uncertainties | Constructs the likelihood distribution shown in eq. (3) for one or multi-bin statistical models. The main model is defined as the product of Poisson distributions across all bins, while the constraint term is defined as the product of unit-Gaussians per nuisance parameter. |
| `"default_pdf.correlated_background"`, Arguments: signal yields, background yields, data, covariance matrix. | Expands upon the uncorrelated background prescription by representing the constraint term using a multivariate Gaussian with the correlation matrix between bins. See eq. (7). |
| `"default_pdf.third_moment_expansion"`, Arguments: signal yields, background yields, data, covariance matrix, diagonal elements of third moments. | Expands the likelihood prescription with skewness information, as presented in ref. [14]. Correlated constraint terms are captured using a multivariate unit-Gaussian distribution. See eq. (8). |
| `"default_pdf.effective_sigma"`, Arguments: signal yields, background yields, data, correlation matrix, upper and lower uncertainty envelops for the background. | Implements asymmetric background uncertainty envelopes using the effective sigma approach, as described in ref. [12]. Correlated constraint terms are captured using a multivariate unit Gaussian. See eq. (10). |
| `"pyhf.uncorrelated_background"`, Arguments: signal yields, background yields, data, background uncertainties. | Constructs a simple likelihood using the uncorrelated background routine from `pyhf`. See section 2.2 for installation details [18]. |
| `"pyhf"`, Arguments: background only description of the likelihood and corresponding signal patch. | Constructs full statistical model within the `pyhf` interface [18]. See section 2.2 for details on installation. |

```
2     signal_yields=[12.0, 15.0],
3     background_yields=[50.0, 48.0],
4     data=[36, 33],
5     absolute_uncertainties=[12.0, 16.0],
6     analysis="example",
7     xsection=0.123,
8   )
```

The `analysis` keyword in line 6 is an optional unique name for the model, and `xsection` in line 7 is the cross-section value of the signal used in the model, with the units determined by the user.[8]

Once initialized, the statistical model inherits all the properties of the `StatisticalModel` class, providing a backend-agnostic interface. The functions provided by the `StatisticalModel` class are summarized in Table 2.

SPEY employs the asymptotic formulae for testing new physics, as outlined in ref. [23]. In

---

[8]Cross-section information is only used for `excluded_cross_section` function where the upper limit on POI is multiplied with the cross-section value provided by the user; hence, the unit is directly propagated without modification. Thus, for SPEY this is a unitless value.

this framework, the test statistic is divided into three main classes: Discovery ($q_0$),

$$q_0 = \begin{cases} 0, & \text{if } \hat{\mu} < 0, \\ -2\log\frac{\mathcal{L}(\mu,\theta_\mu)}{\mathcal{L}(\hat{\mu},\hat{\theta})}, & \text{otherwise}, \end{cases} \tag{4}$$

$q_\mu$ test statistic,

$$q_\mu = \begin{cases} 0, & \text{if } \hat{\mu} > \mu, \\ -2\log\frac{\mathcal{L}(\mu,\theta_\mu)}{\mathcal{L}(\hat{\mu},\hat{\theta})}, & \text{otherwise}, \end{cases} \tag{5}$$

and alternative test statistic $\tilde{q}_\mu$,

$$\tilde{q}_\mu = \begin{cases} 0, & \text{if } \hat{\mu} > \mu, \\ -2\log\frac{\mathcal{L}(\mu,\theta_\mu)}{\mathcal{L}(\hat{\mu},\hat{\theta})}, & \text{if } 0 < \hat{\mu} < \mu, \\ -2\log\frac{\mathcal{L}(\mu,\theta_\mu)}{\mathcal{L}(0,\theta_0)}, & \text{otherwise}, \end{cases} \tag{6}$$

for upper limits. Each test statistic covers a specific domain defined by the interplay between the profiled likelihood, $\mathcal{L}(\mu,\theta_\mu)$, background only profiled likelihood, $\mathcal{L}(0,\theta_0)$, and the maximum likelihood values, $\mathcal{L}(\hat{\mu},\hat{\theta})$. Here $\hat{\mu}$ and $\hat{\theta}$ are the set of parameters that maximises the likelihood, and $\theta_\mu$ are the nuisance parameters that maximises the likelihood for a specific POI.

For computing p-values with respect to the desired test statistic, SPEY offers two methods: using asymptotic formulae or generating samples from the PDF distribution. In the aforementioned example, the exclusion confidence level, $1 - \mathrm{CL_s}$, can be computed using the `statistical_model.exclusion_confidence_level` function, with $\tilde{q}_\mu$ test statistic, yields a value of 97% CL.

SPEY offers three distinct approaches for computing limits, which are designated by the `expected` keyword. This keyword can be set using `spey.ExpectationType.<XXX>`, where `<XXX>` can take on the values `observed`, `aposteriori`, or `apriori`.

The `observed` and `aposteriori` expectation types involve conducting a likelihood fit using the provided observations during the initialisation of the statistical model. The distinction between these two types becomes evident when computing the p-values.

For the `observed` expectation type, the functions `statistical_model.exclusion_confidence_level` and `statistical_model.poi_upper_limit` provide observed results. In this scenario, the cumulative distribution function (CDF) is computed using two asymptotic distributions: the background-only distribution ($\mathrm{CL}_b$) and the signal-plus-background distribution ($\mathrm{CL_{s+b}}$). The computation involves comparing the test statistic calculated using the observations to the data generated by the background-only statistical model, also known as Asimov data. The observed p-value is obtained by evaluating the ratio $\mathrm{CL_s} = \mathrm{CL_{s+b}}/\mathrm{CL_b}$ (for more details, see ref. [23]).

For the `aposteriori` expectation type, one and two sigma fluctuations around the background-only distribution are calculated.

On the other hand, the `apriori` approach involves fitting the expected background yields, assuming the Standard Model as the truth, and subsequently computing expected p-values, similar to `aposteriori` expectation type. This option is commonly utilised by theorists, such as in the expected results presented in MADANALYSIS 5 [24] and SModelS [25, 26], where fitting is performed based on expected background yields.

To illustrate the distinction between the `observed` and `apriori` expectation types, one can plot the likelihood distributions using the `statistical_model.likelihood()` function. This will immediately reveal a shifted likelihood distribution: for the `apriori` case, the profiled likelihood peaks at zero ($\hat{\mu} = 0$), whereas for the `observed` case, it peaks at $\hat{\mu} = -1.08$, computed

Table 2: Functions provided by the `StatisticalModel` class.

| Functions and Properties | Functionality |
| --- | --- |
| `exclusion_confidence_level()` | Computes the exclusion confidence level $(1-CL_s)$ for an observed or expected results. Can be computed for eq. (5) or eq. (6). |
| `excluded_cross_section()` | Computes the upper limit on the cross-section, with the unit depending on the input unit of `xsection`. |
| `likelihood()` | Computes the profiled likelihood for a given POI. |
| `maximize_likelihood()` | Computes the maximum likelihood by fitting the likelihood to observed data or background yields. |
| `generate_asimov_data()` | Generates Asimov data. |
| `asimov_likelihood()` | Computes the profiled likelihood for a given POI by fitting the likelihood to the Asimov data. |
| `maximize_asimov_likelihood()` | Computes the maximum likelihood by fitting the likelihood to the Asimov data. |
| `poi_upper_limit()` | Computes the upper limit on the parameter of interest (POI). Can be computed for eq. (5) or eq. (6). |
| `sigma_mu()` | Computes the variance on the POI using Asimov likelihood, where $\sigma^2\mu \approx \mu^2/q_{\mu,A}$ (see ref. [23] for details). |
| `sigma_mu_from_hessian()` | Computes the variance on the POI via the Hessian of the negative log-likelihood. |
| `significance()` | Computes the significance of the signal yields using eq. (4). |
| `fixed_poi_sampler()` | Samples from the statistical model by profiling the likelihood with a fixed POI. |
| `chi2()` | Computes the likelihood ratio of the profiled likelihood at a fixed POI to the maximum likelihood, i.e., $\chi^2(\mu) = -2\log\left(\mathcal{L}\left(\mu,\theta_\mu\right)/\mathcal{L}\left(\hat{\mu},\hat{\theta}\right)\right)$. |
| `combine()` | If a backend-specific combination routine has been implemented, it combines two statistical models. |
| `available_calculators` | Retrieves information on which calculator is available for upper limit computations, i.e., asymptotic or toy-based. |

via the `statistical_model.maximize_likelihood()` function. This difference diminishes as the observed data approaches the background yields.

Utilizing the same statistical model, the `significance` function is employed to assess the discovery significance of this model. By utilizing eq. (4), the function computes significance ($Z$), represented as $\sqrt{q_{0,A}} = 5 \times 10^{-4}$, alongside determining $\sqrt{q_0}$ and expected and observed p-values relative to eq. (4). Consistent with the earlier presented exclusion limit, the significance of this signal yield is notably low.

## 2.1 Correlated histograms with simplified likelihoods

Under the assumption that the uncertainties are modelled as Gaussian distributions, one can employ several methods to incorporate this information into PDF distribution. The simplest approach would be extending the constraint term with the correlation matrix (or covariance matrix) provided by the experiment,

$$\mathcal{L}(\mu,\theta) = \left[\prod_{i \in \text{bins}} \text{Poiss}\left(n^i | \mu n_s^i + n_b^i + \theta^i \sigma_b^i\right)\right] \cdot \mathcal{N}\left(\theta | 0, \rho\right), \qquad (7)$$

where $\rho$ represents the correlation matrix between each nuisance parameter. In such a simplified scenario, various uncertainty sources are contracted into a single uncertainty for each histogram bin. This PDF can be accessed via `"default_pdf.correlated_background"`.

Despite its efficiency, such models do not capture asymmetric uncertainties, which skew the likelihood distribution. In order to address this issue, ref. [14] proposed an expansion procedure to eq. (7) via the third moments of the background uncertainty where likelihood distribution has been expanded as

$$\mathcal{L}(\mu, \theta) = \left[ \prod_{i \in \text{bins}} \text{Poiss}\left(n^i | \mu n_s^i + \bar{n}_b^i + A_i \theta_i + C_i \theta_i^2\right) \right] \cdot \mathcal{N}(\theta | 0, \bar{\rho}) . \tag{8}$$

Here $\bar{n}_b$ are the central values of the expected background yields, $A$ are the effective sigma within the symmetric covariance matrix approximation, and $C$ represents the skewness of the distribution, which are given as;

$$C_i = -\text{sign}(m_i^{(3)}) \sqrt{2 \, \text{diag}(\Sigma)_i^2} \times \cos\left( \frac{4\pi}{3} + \frac{1}{3} \arctan\left( \sqrt{\frac{8(\text{diag}(\Sigma)_i^2)^3}{(m_i^{(3)})^2} - 1} \right) \right),$$

$$A_i = \sqrt{\text{diag}(\Sigma)_i - 2C_i^2}, \tag{9}$$

$$\bar{n}_b^i = n_b^i - C_i,$$

$$\bar{\rho}_{ij} = \frac{1}{4C_i C_j} \left( \sqrt{(A_i A_j)^2 + 8C_i C_j \Sigma_{ij}} - A_i A_j \right),$$

where $m^{(3)}$ are the diagonal elements of the third moments and $\Sigma$ is the covariance matrix. Notice that $A$ and $C$ also modifies the correlation matrix, $\bar{\rho}$, which enters the eq. (8). Such expansion provides a more accurate representation of the original likelihood distribution by integrating the skewness into the PDF. This PDF can be accessed via `"default_pdf.third_moment_expansion"` accessor.[9] Notice that when skewness is zero, $C = 0$, the expansion reduces to eq. (7).

The skewness of the PDF distribution can also be captured by building an effective variance from the upper $(\sigma^+)$ and lower $(\sigma^-)$ uncertainty envelops as a function of nuisance parameters,

$$\sigma_{\text{eff}}^i(\theta^i) = \sqrt{\sigma_i^+ \sigma_i^- + (\sigma_i^+ - \sigma_i^-)(\theta^i - n_b^i)} .$$

This method has been proposed in ref. [12] for Gaussian models which can be generalised for eq. (7) by reparametrising the likelihood distribution,

$$\mathcal{L}(\mu, \theta) = \left[ \prod_{i \in \text{bins}} \text{Poiss}\left(n^i | \mu n_s^i + n_b^i + \theta^i \sigma_{\text{eff}}^i(\theta^i)\right) \right] \cdot \mathcal{N}(\theta | 0, \rho) , \tag{10}$$

effectively implementing the skewness into the Poisson distribution.[10] This PDF can be accessed via `"default_pdf.effective_sigma"` accessor.[11] Notice that if $\sigma^+ = \sigma^-$, eq. (10) reduces to eq. (7).

---

[9]The implementation of the third moment expansion has been validated against the code provided by ref. [14] which can be found in the dedicated GitLab repository.

[10]In the context of reinterpretation, this method has also been employed by ref. [27], without the Poisson term, as defined in ref. [12], i.e. variable Gaussian.

[11]It is also possible to incorporate variable Gaussian directly into the constraint term as presented in ref. [12]; however, we observed that the optimisation landscape becomes highly complicated to ensure a reliable outcome.

Going beyond background-only uncertainties, eq. (3) can be expanded to accommodate signal uncertainties and employ the aforementioned methodologies. To achieve this, the constraint term has been extended with a signal-specific constraint term

$$\mathcal{C}(\theta) \subset \mathcal{N}^{\mu}(\theta_s | 0, \rho_s),\tag{11}$$

where $\theta_s$ are the nuisance parameters dedicated for signal uncertainties and $\rho_s$ is the correlation matrix between nuisance parameters. In case $\rho_s$ is not known, constraint terms reduces to $\mathcal{N}^{\mu}(\theta_s | 0, 1)$, as before. Notice that the constraint term has been scaled with POI, which is necessary to capture the scaling on signal yields. Similarly, depending on the information provided, $\lambda_i(\mu, \theta)$ can be modified with third-moment expansion or effective sigma. Signal uncertainties have been integrated via `signal_uncertainty_configuration` keyword for each `default_pdf` backend. Note that this implementation assumes signal uncertainties are not correlated with background uncertainties.

### 2.1.1 Comparing simplified approaches

To test the accuracy of these frameworks presented above against a realistic experimental analysis, we used CMS-SUS-20-004 [28], which searches for four jet and large missing energy phase-space at a centre-of-mass energy of 13 TeV and 137 fb$^{-1}$ integrated luminosity. It focuses on the production of two Higgs decaying into $b\bar{b}$ alongside with two lightest neutralino, $\tilde{\chi}_1^0$ through two degenerate heavy neutralino mediators, $\tilde{\chi}_{2,3}^0$,

$$pp \rightarrow \tilde{\chi}_2^0 \tilde{\chi}_3^0 \rightarrow H(\rightarrow b\bar{b})H(\rightarrow b\bar{b})\,\tilde{\chi}_1^0 \tilde{\chi}_1^0.$$

The analysis provides yields, covariance matrix and asymmetric uncertainties for 22 signal regions which can be found from the dedicated HEPData entry [29].

Utilising the signal yields provided by the collaboration, we constructed three different statistical models within SPEY, referring to correlated backgrounds, third-moment expansion (we refer the reader to Appendix C for the computation of third moments) and a model with effective sigma method (eq. (10)). Fig. 1 shows the expected limits within heavy ($\tilde{\chi}_{2,3}^0$) and light ($\tilde{\chi}_1^0$) neutralino mass plane, provided by these three likelihood prescriptions where the black curve is the original CMS expected limit within $\pm 1\sigma$ window, presented with dotted lines. Red, blue and green curves represent the result computed with correlated background, third-moment expansion and effective sigma model. As before, dotted lines represent each curve's $\pm 1\sigma$ window.

As can be seen, for a large number of signal yields, $m_{\tilde{\chi}_{2,3}^0} < 450$ GeV, all likelihood prescriptions provide accurate results in terms of reproducing the exclusion curve supplied by CMS. However, for the regions with low signal yields, $m_{\tilde{\chi}_{2,3}^0} > 450$ GeV, the correlated background approach overestimates the exclusion limit by approximately 80 GeV. The model with third-moment expansion reduces this difference to about 40 GeV. On the other hand, the model with effective sigma slightly underestimates the expected exclusion limit by only a few GeV. Similarly, this approach reproduces the uncertainty bands much more accurately than the other two for a low number of signal yields.

To have a closer look, we have chosen a point in Fig. 1 where correlated background and third-moment expansion is the closest to the CMS limit where the effective sigma model is further away. Using the $\chi^2$ distribution for the full statistical model, provided by the CMS collaboration (see the corresponding HEPData entry), we chose $m_{\tilde{\chi}_{2,3}^0} = 300$ GeV, $m_{\tilde{\chi}_1^0} = 50$ GeV point to compare all three PDF prescription against the full statistical model where the $\chi^2$ distribution is given as,

$$\chi^2(\mu) = -2\log\left(\frac{\mathcal{L}(\mu, \theta_\mu)}{\mathcal{L}(\hat{\mu}, \hat{\theta})}\right).\tag{12}$$

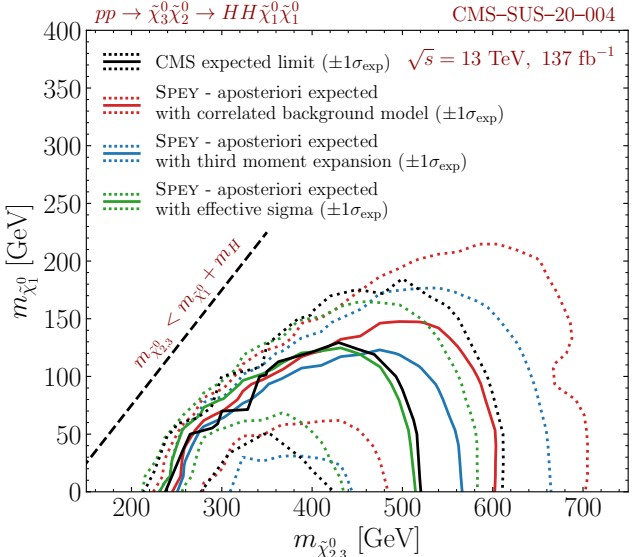

Figure 1: *Expected exclusion limits at 95% CL presented for CMS-SUS-20-004 analysis. Black, red, blue and green curves represent CMS expected limit, and expected limits computed using correlated background model (eq. (7)), third-moment expansion (eq. (8)) and effective sigma (eq. (10)) methods, respectively. The dotted lines for each curve represent $\pm 1\sigma$ fluctuation from the background. The dashed black line is plotted as a reference where $m_{\tilde{\chi}_{2,3}^0}$ becomes lighter than the mass combination of Higgs and the lightest neutralino.*

Here denominator shows the maximum likelihood value where $\hat{\mu}, \hat{\theta}$ are the values that maximize the likelihood, and the nominator shows the profiled likelihood for a given $\mu$ where $\theta_\mu$ are the nuisance parameters that maximize the likelihood at a given $\mu$. This value can be computed via `chi2` function as shown in Table 2. Fig. 2 shows the $\chi^2$ distribution for all three curves compared against CMS full statistical model presented as black. The red, blue and green lines represent correlated background, third-moment expansion and effective sigma models. We additionally provided observed POI upper limits for each model, which are colour coded where the effective sigma model underestimates $\mu_{95\%CL}^{\mathrm{obs}}$ by 18% and third-moment expansion overestimates it by only 3%. This value can be computed via `poi_upper_limit` function as shown in Table 2. The upper limit on the expected cross section at this mass grid has been given as 36.06 fb where correlated background, third-moment expansion and effective sigma models set this limit to $38.32^{+15.15}_{-10.07}$ fb, $41.66^{+14.95}_{-10.02}$ fb and $34.11^{+13.09}_{-8.74}$ fb, respectively where each value is computed with $1\sigma$ deviation.

Asymmetric background uncertainties have also been provided by CMS-SUS-19-006 analysis [30] through its dedicated HEPData record [31] which is conducted at 13 TeV centre-of-mass energy with 137 fb$^{-1}$ luminosity. This analysis is designed to investigate new physics signatures through multi-jet and missing energy final states through gluino production, which consequently decays into a $t\bar{t}$ pair and lightest neutralino,

$$pp \to \tilde{g}(\to t\bar{t}\tilde{\chi}_1^0)\,\tilde{g}(\to t\bar{t}\tilde{\chi}_1^0).$$

The analysis includes 174 non-overlapping signal regions[12] classified with respect to jet and b-jet multiplicity, hadronic transverse momentum and missing energy. The events are required to have minimum 300 GeV hadronic activity ($H_T$) and $H_T^{miss}$. At least two jets have been

---

[12]Aggregated regions are not considered in this study since the correlation matrix does not include those regions.

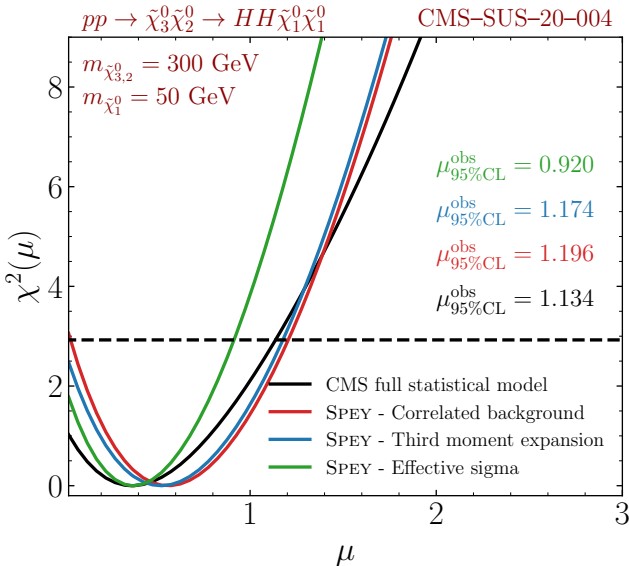

Figure 2: $\chi^2(\mu)$ *distribution versus parameter of interest shown for CMS full statistical model (black), correlated background (red), third-moment expansion (blue) and effective sigma (green). The dashed black line shows the $\chi^2$ value that CMS excludes, and colour-coded values represent excluded POI value at 95% CL by each PDF.*

required in each bin with $p_T > 30$ GeV. Additionally, events including isolated high $p_T$ leptons or photons are removed.

For this analysis, we used MADANALYSIS 5 implementation [32] where the hard scattering process, $pp \rightarrow \tilde{g}\tilde{g}$, has been simulated via MADGRAPH5_AMC@NLO version 3.2.0 with MSSM_SLHA2 UFO model [33] where BR($\tilde{g} \rightarrow t\bar{t}\tilde{\chi}_1^0$) has been set to 100%. We used the leading order set of NNPDF 2.3 parton distribution function [34,35] and showered hard scattering events via PYTHIA version 8.240 [36]. Finally, the leading-order cross section has been scaled to next-to-next-to-leading-logarithmic accuracy [37]. To perform detector simulation, the recast employs Delphes, for which we used version 3.5 [38].

Fig. 3 shows the comparison between the official exclusion limit (black), `"default_pdf.correlated_background"` (red), `"default_pdf.third_moment_expansion"` (blue) and `"default_pdf.effective_sigma"` (green) models. Provided uncertainties are separated into systematic and statistical uncertainties, which we combined in quadrature. Note that we scaled the third moments, computed by eq. (C.1), by 0.9 to achieve numeric stability and satisfy the constraints in eq. (9). The left panel presents a comparison between observed exclusion limits in the ($m_{\tilde{g}}$, $m_{\tilde{\chi}_1^0}$) plane. However, we only observe a small improvement over `"default_pdf.correlated_background"` for both expansions. Conversely, the right panel compares the aposteriori (`spey.ExpectationType.aposteriori`) expectation limit produced by each model. We observe over 50 GeV improvement for the expected exclusion limit when `"default_pdf.effective_sigma"` model is used, and a minor but noticeable refinement by `"default_pdf.third_moment_expansion"`. Although we still observed slight over-exclusion, both models produce significantly more accurate results than the correlated background approach.[13]

Although the effective sigma method managed to reproduce the exclusion curve in Figures 1 and 3 better than the other approaches for low signal yield regions, it is essential to emphasise that this is only an approximation; hence its case-dependent. As shown in Fig. 2, it can

---

[13]One can observe significant differences between the results presented in Fig. 3 and ref. [24], this is due to various important bug fixes in the implementation and the usage of aposteriori versus apriori expectation limits.



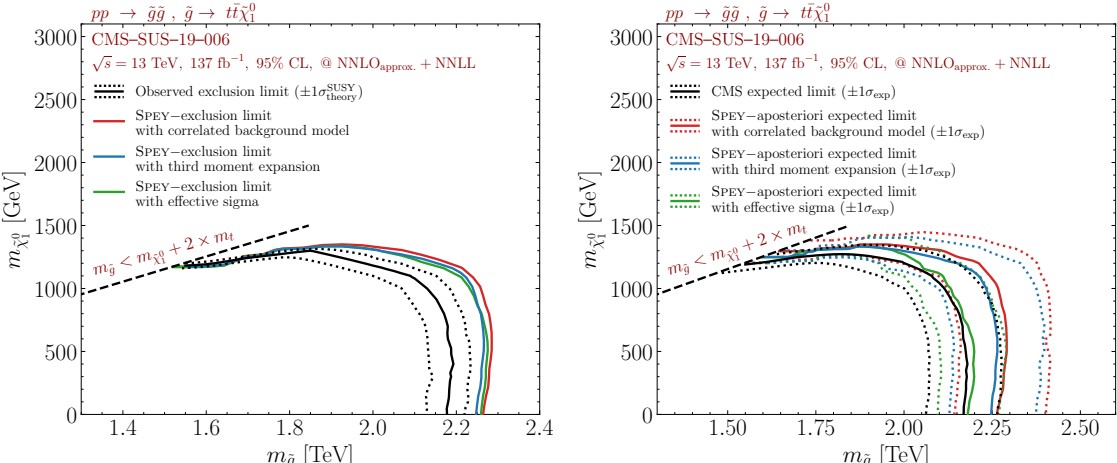

Figure 3: *95% CL exclusion contours for the process $pp \rightarrow \tilde{g}\tilde{g}$, $\tilde{g} \rightarrow t\bar{t}\tilde{\chi}_1^0$ in the $(m_{\tilde{g}}, m_{\tilde{\chi}_1^0})$ plane using the* MADANALYSIS *5 recast of the CMS-SUS-19-006 analysis. The left panel shows observed limits, and the right panel shows the expected limits with one standard deviation from the background hypothesis. The official limits from the CMS collaboration have been shown with black curves for each plot. The exclusion limits that are obtained by* `"default_pdf.correlated_background"`, `"default_pdf.third_moment_expansion"` *and* `"default_pdf.effective_sigma"` *models have been shown in red, blue and green, respectively.*

vastly underestimate the exclusion limit even when signal yield is adequate. PDF distributions with a larger third moment or an assumption beyond purely Gaussian uncertainties might be more suitable to exclude low signal yield regions. Thus, one should always be aware of the limitations of each approach and test the results accordingly.

## 2.2 Full statistical models

As mentioned in Sec. 2, SPEY has been built to be expandable to exploit other packages that are designed to provide specific PDF prescriptions. `pyhf` [18] is a Python package designed based on `HistFactory` [39], which allows publication and usage of full statistical models through a JSON sterilised format.

The `spey-pyhf` plug-in enables SPEY to exploit the PDF constructed by `pyhf` interface and describe it within `StatisticalModel` class. This provides a completely backend-agnostic interface where no matter the PDF function's origin, it can be executed through the same functions, presented in Table 2. `spey-pyhf` plug-in can be installed via

```
1  pip install spey-pyhf
```

command,[14] and once installed, SPEY can automatically detect it. The bottom section of Table 1 shows available accessors for `pyhf` plug-in where `"pyhf"` accessor will allow the user to input full statistical model prescriptions as defined in `pyhf`'s online manual. Uncorrelated background samples can also be studied using `"pyhf.uncorrelated_background"` accessor. As shown before, the `spey.get_backend` function will enable the usage of these accessors.[15]

The usage of a full statistical model can be demonstrated using ATLAS-SUSY-2018-31 [40] analysis where we used the recast of the study implemented within MADANALYSIS 5 [41, 42]

---

[14]Dedicated online documentation can be found in this link.

[15]Certain functionalities of `pyhf` package are limited to the choice of its backend. The gradient of the negative log-likelihood function is only available through Jax or Tensorflow, and the computation of the variance via `sigma_mu_from_hessian` function is currently only available with `pyhf`'s Jax backend.

package through SFS interface [9, 43].[16] This analysis is designed to search for new physics for multi-jet final state scenarios where it's looking for sbottom production with

$$pp \to \tilde{b}_1(\to b\tilde{\chi}_2^0) \, \tilde{b}_1^*(\to b\tilde{\chi}_2^0), \; \tilde{\chi}_2^0 \to bH\tilde{\chi}_1^0 \, ,$$

decay pattern where $\tilde{b}_1$ and $\tilde{\chi}_2^0$ decay branching ratios are set to 100%. The analysis includes eight signal regions grouped into three super regions called A, B and C, designed to capture different levels of mass spectra compression. The HEPData entries for this analysis can be found in ref. [44].

The hard scattering process, $pp \to \tilde{b}_1\tilde{b}_1$, has been simulated with MADGRAPH5_AMC@NLO version 3.2.0 [45] using MSSM_SLHA2 model [33]. The leading order set of NNPDF 2.3 has been used as a parton distribution function [34, 35], and events are showered as well as decayed via PYTHIA version 8.240 [36]. The leading-order signal cross section has been scaled to approximate next-to-next-to-leading-order matched with soft-gluon resummation at the next-to-next-to-leading-logarithmic accuracy [37].

The left panel of Fig. 4 shows the observed exclusion limits in $(m_{\tilde{b}_1}, \, m_{\tilde{\chi}_2^0})$ plane where official limits published by ATLAS have been shown in blue. This analysis has been shipped with three different background-only statistical model descriptions allowing three distinct subsets of the signal regions to be used to create different statistical models. For the red curve, we choose the most sensitive statistical model for each mass grid by selecting the statistical model that produces the lowest POI upper limit at 95% CL. For $m_{\tilde{\chi}_2^0} < 1$ TeV, we observed that the best statistical model is dominated by so-called region A, which requires the highest jet and b-tagged jet multiplicity along with boosted leading b-jet. Since it is possible to combine statistical models by matching their modifiers, we combined all three statistical models for the orange curve. This has been achieved by `combine()` function implemented through `StatisticalModel` class. This function employs the workspace combiner routine implemented in `pyhf` interface and modifies the signal input to be accommodated into the new background-only model. We observe that the difference between red and orange curves is only visible when the difference between POI upper limits for each subregion (regions A, B and C) is significantly closer to each other.

The right panel of Fig. 4 shows the same for expected exclusion limits along with one standard deviation from the background hypothesis, displayed with dotted lines. We observed the exact difference between the red and orange curves once the phase space is not statistically dominated by one sub-region. This deviation from the red curve has been observed to fit the official limits better in the region where the mass spectra are highly compressed.

Whilst such a combination enables one to merge multiple full statistical models by properly taking care of the common nuisance parameters, it only applies to a full likelihood scenario where all the nuisance parameters are properly identified. Needless to say, this assumes that there is a fixed naming convention in place within the experimental collaboration. In the next section, we will discuss the possibility of combining statistical models which do not include complete information but have been formed using approximation techniques discussed in Sec. 2.1.

## 3  Combination of statistical models

The versatile modular interface of SPEY facilitates the comprehensive study of various PDF descriptions within a single package. This advantageous feature empowers researchers to employ

---

[16]Our implementation has been validated against ref. [24]. Validation note of the recast can be found in this link.

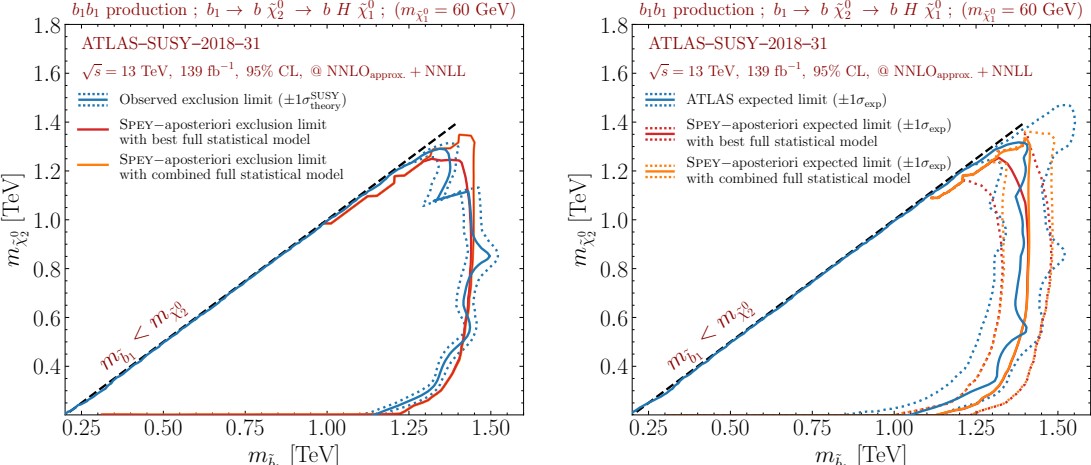

Figure 4: *95% CL exclusion contours for the process $pp \rightarrow \tilde{b}_1\tilde{b}_1, \tilde{b}_1 \rightarrow bH\tilde{\chi}_1^0$ at $m_{\tilde{\chi}_1^0} = 60$ GeV plotted in the ($m_{\tilde{b}_1}$, $m_{\tilde{\chi}_2^0}$) plane using the* MADANALYSIS *5 recast of ATLAS-SUSY-2018-31 analysis. The left panel shows observed limits, and the right shows the expected limits with one standard deviation from the background model. The official limits from the ATLAS collaboration have been shown with blue curves for each plot. The exclusion limits were obtained via* `"pyhf"` *plug-in. The red curve shows the most sensitive statistical model among three different background-only models shipped with this analysis. The orange curve shows the results produced by combining these three models.*

various PDF combination methodologies, thereby harnessing the full potential of experimental analyses. Specifically, the combination methodology outlined in Sec. 2.2 is applicable when complete knowledge of uncertainty sources enables the matching of nuisance parameters associated with the same uncertainties. Consequently, this statistical model expansion introduces a correlation term that characterizes any existing interdependencies among nuisance parameters. A generic combination of different statistical models can be formulated as

$$\mathcal{L}'(\mu, \theta) = \prod_{i \in \text{models}} \mathcal{L}_i(\mu, \theta_i) \cdot \prod_{i,j \in \text{nui}} \mathcal{C}_{i,j}(\theta_i, \theta_j). \qquad (13)$$

Here, $i$ represents the statistical models, and $\theta_i$ corresponds to the independent nuisance parameters associated with each model. The combined likelihood in eq. (13) encompasses two key components. The first term involves multiplying the constituent PDFs obtained from different statistical models, while the second term introduces constraints that account for correlations among the associated nuisance parameters. Common nuisance parameters have been identified and appropriately addressed to avoid double counting. However, in the case of independent experiments, it is reasonable to assume negligible correlations between the nuisance parameters. This assumption leads to a simplified form of the likelihood, expressed as:

$$\mathcal{L}'_{\text{indep.}}(\mu) = \prod_{i \in \text{models}} \mathcal{L}_i(\mu, \theta_i). \qquad (14)$$

Notably, the combined likelihood solely relies on the parameter of interest (POI) since each statistical model can be independently profiled with respect to the given POI value. It is crucial to emphasize that constructing a constraint term between two experiments is highly challenging due to limited information about the sources of uncertainties and overlapping regions. Furthermore, even if the sources of uncertainties are known, such as the jet energy scale or muon

tracking efficiencies, the techniques employed to quantify these uncertainties may vary over time or between different collaborations. This variation leads to uncorrelated uncertainties, further validating the validity of eq. (14).[17]

An alternative approach worth exploring is the combination of different non-overlapping regions, as demonstrated in [20]. This technique eliminates the correlations between statistical models, further bolstering the validity of equation (14). However, it is important to note that this approach has some limitations. The construction of the overlap matrix is based on the phase space populated by signal events, and thus it may not fully capture the effects of background contributions. Additionally, the success of this method heavily relies on the recasting procedure, as control and validation regions are typically excluded from the recast. Consequently, the algorithm remains blind to those regions, potentially resulting in overlaps between signal regions in one analysis and control or validation regions in another analysis. Despite these limitations, this approach offers a semi-conservative strategy for combining regions and analyses when likelihood information is significantly constrained.

To evaluate the impact of combining likelihoods, we adopt a realistic MSSM scenario as outlined in ref. [24]. In this scenario, the production of the lightest sbottoms and stops,

$$pp \rightarrow \tilde{b}_1 \bar{\tilde{b}}_1 \text{ and } \tilde{t}_1 \bar{\tilde{t}}_1 ,$$

typically leads to decays involving charginos, top and bottom quarks, respectively. These decays give rise to a final state with multiple jets and significant missing energy due to the production of a bino-like lightest neutralino through chargino interactions. ATLAS-SUSY-2018-31 and CMS-SUS-19-006 analyses are ideal for this particular final state configuration, which is discussed above.

To ensure the validity of the theory on the electroweakino side, we additionally employed a comprehensive electroweakino production channel,

$$pp \rightarrow \tilde{\chi}_1^\pm \tilde{\chi}_2^0 , \ \tilde{\chi}_1^\pm \tilde{\chi}_1^\mp , \text{ and } \tilde{\chi}_2^0 \tilde{\chi}_2^0 , \tag{15}$$

using the same configurations mentioned above. We used the MADANALYSIS 5 recasts of ATLAS-SUSY-2018-32 [47, 48],[18] ATLAS-SUSY-2019-08 [49, 50], and CMS-SUS-16-039 [51, 52] to study the exclusion limits on the MSSM scenario, which revealed a lower limit of $M_2 > 650$ GeV for our entire analysis. This lower limit will be included in the below results as a shaded area.

The hard scattering process for both squark and electriweakino production has been simulated with MADGRAPH5_AMC@NLO version 3.2.0 [45] using MSSM_SLHA2 model [33]. All model parameters, masses, and decay widths are computed using the SoftSusy package [53, 54]. The leading order set of NNPDF 2.3 has been used as a parton distribution function [34, 35]. Finally, events are showered and decayed via PYTHIA version 8.240 [36].

We follow the same grid scan employed in the reference, where the parameters $\tilde{b}_1$, $\tilde{t}_1$, $\tilde{\chi}_1^\pm$, and $\tilde{\chi}_{1,2}^0$ are allowed to vary within the sub-TeV range. At the same time, the other supersymmetric partners have heavier masses. To achieve this, we fix the bino mass ($M_1$) at 60 GeV and set the ratio of the Higgs vacuum expectation values ($\tan \beta = v_2/v_1$) to 10. The trilinear couplings ($A_{t,b}$), $\mu$, and other soft masses are assigned values of $-3.5$ TeV, 1.6 TeV, and 5 TeV, respectively. The remaining parameters are scanned in the order $2M_1 < M_2 < M_{\tilde{Q}_3}$. This arrangement ensures that the sbottom and stop masses are approximately equal to $M_{\tilde{Q}_3}$, the lightest chargino and the second lightest neutralino have a mass equal to $M_2$, and the bino-like lightest neutralino has a mass fixed at 60 GeV.

---

[17]For a comprehensive discussion on likelihood combinations, we recommend referring to ref. [46].

[18]The metadata has been updated during this study to include full statistical model information.

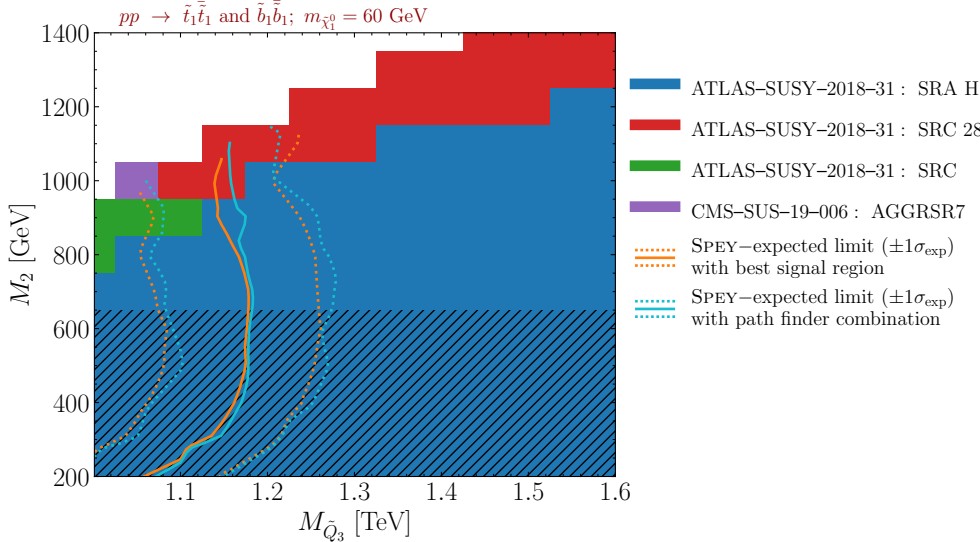

Figure 5: *95% confidence level exclusion limits on $pp \to \tilde{b}_1 \bar{\tilde{b}}_1$ and $\tilde{t}_1 \bar{\tilde{t}}_1$ production channels in MSSM scenario presented in $(M_{\tilde{Q}_3}, M_2)$ mass plane. The orange line represents the exclusion limit computed by using the most sensitive signal region, and the cyan line is computed by combining signal regions via the pathfinder algorithm. Other colours represent the most sensitive signal region per grid point. The hashed area is excluded by the electroweakino searches.*

For the recasts of ATLAS-SUSY-2018-31 and CMS-SUS-19-006, we utilized the same configurations as described previously. Specifically, we generated a dataset comprising 200,000 events for the $pp \to \tilde{b}_1 \bar{\tilde{b}}_1$ and $\tilde{t}_1 \bar{\tilde{t}}_1$ production channels.

Figure 5 illustrates the anticipated limits and identifies the most sensitive signal regions established using the `"default_pdf.uncorrelated_background"` model. Determining the most sensitive region was based on the statistical model that yielded the lowest expected upper limit on the parameter of interest. Notably, our analysis reveals the prominence of four distinct regions.

In the ATLAS-SUSY-2018-31 analysis, the SRA (Signal Region A) is characterized by specific selection criteria. Firstly, it necessitates a jet multiplicity exceeding 6 and a minimum of 4 b-tagged jets. Moreover, the transverse momentum of a b-tagged jet is expected to surpass 200 GeV. To accommodate the Higgs mass, a restriction on the $\Delta R$ separation between two b-jets has been imposed, alongside a loose mass requirement of 80 GeV.

Additionally, the H variant of the SRA requires the highest effective mass in the analysis, surpassing 2 TeV. Notably, within the blue mass grid region, there exists a greater mass separation between squarks and charginos, which consequently leads to an increased multiplicity of boosted jets in this particular region.

In the ATLAS-SUSY-2018-31 analysis, the SRC (Signal Region C) imposes a lower jet multiplicity requirement compared to SRA. Specifically, it necessitates a minimum of 4 jets and 3 B-tagged jets. Additionally, SRC includes a minimum missing energy threshold of 250 GeV. Notably, SRC 28, which is a specific region within SRC, further demands a high missing energy significance of $E_T^{\mathrm{miss}}/\sqrt{H_T} > 28 \sqrt{\mathrm{GeV}}$. These two regions, SRC and SRC 28, have been observed to dominate in scenarios with compressed spectra. As the squark mass increases, the significance of high missing energy also becomes more pronounced.

Furthermore, within the context of the CMS-SUS-19-006 analysis, only a single grid point is observed. This region specifically targets a low jet multiplicity requirement ($N_j \geq 2$ and

$N_b \geq 2$) while also ensuring significant hadronic activity ($H_T$) and a minimum level of missing hadronic activity ($H_T^{\text{miss}} > 600$ GeV).

In Fig. 5, the orange curve represents the expected exclusion limit, highlighting the most sensitive region. The dotted lines indicate the $\pm 1\sigma$ deviation from the background hypothesis. To perform the combination of likelihoods, we employed the pathfinder algorithm proposed in ref. [20], visualized with the cyan color.[19]

To construct the overlap matrix, we utilized information regarding the regions populated by specific events generated by MADANALYSIS 5. Each region was weighted by

$$w_i = -\log \frac{\mathcal{L}_i(1, \theta_1)}{\mathcal{L}_i(0, \theta_0)}.$$

These weights for each region can be computed using the `likelihood()` function within the `StatisticalModel` class. Once the statistical models to be combined are determined, the `UnCorrStatisticsCombiner` class can be utilized to combine them. It accepts `StatisticalModel` as input, leaving the validation of the combination to the user's discretion. It is worth noting that the `StatisticalModel.combine()` function is designed for backends with specific combination algorithms, whereas the `UnCorrStatisticsCombiner` does not interact with the statistical models directly but rather multiplies the PDFs as shown in equation (14). It provides the same functionality as the `StatisticalModel`, and once initialized, it can be used as an independent statistical model for computing exclusion and upper limits.

Despite the ongoing dominance of the ATLAS-SUSY-2018-31 analysis in the most sensitive regions, we observed a slight improvement in the cyan curve due to the combination of regions from both analyses. This improvement primarily stems from the inclusion of ATLAS-SUSY-2018-31 SRA-H and two floating regions from the CMS-SUS-19-006 analysis, which fluctuates for different grid points due to insufficient yields. The reason for this modest enhancement over the orange curve lies in the fact that the combined signal regions do not exhibit close $\mu_{95\%CL}$ values. In particular, the $\mu_{95\%CL}$ values for different regions are maximally close in the compressed spectra region. Thus, the combination of these regions contributes to the observed improvement over the baseline orange curve. This means that the contributions from different signal regions only affect if their sensitivity is comparable.

To leverage the statistical model information provided by the analyses, we generated Figure 6, presenting the observed exclusion limits on the left panel and the expected exclusion limits on the right panel. In both panels, the ATLAS-SUSY-2018-31 analysis employed the `"pyhf"` model, indicated by the green curve which includes combination of all three super-regions A, B and C. For the CMS-SUS-19-006 analysis, we utilized the `"default_pdf.effective_sigma"` model as discussed in Section 2.1, represented by the blue curve.

Assuming complete independence between the two experiments, we combined their statistical models using equation (14). This combination was achieved by inputting both models into the `UnCorrStatisticsCombiner`, which combines them regardless of their backend. The combined analysis is shown as the red curve in Figure 6. However, it is important to note that complete independence may not hold in practice. Although the experiments are distinct, they might share uncertainties such as jet energy scaling or luminosity. Consequently, this combination should be regarded as the maximum information that can be extracted from these analyses based on the available statistical model information. Any correlations between the analyses would reduce the exclusion limit.

Furthermore, the range covered by these analyses differs significantly due to requirements in terms of multiplicity and transverse energy. We observe an improvement of up to 100 GeV with this combination, particularly when constituent regions yield similar exclusion limits. Thus, the red curve represents a highly conservative exclusion limit by utilizing both analyses.

---

[19]The pathfinder algorithm can be accessed from this GitHub repository.

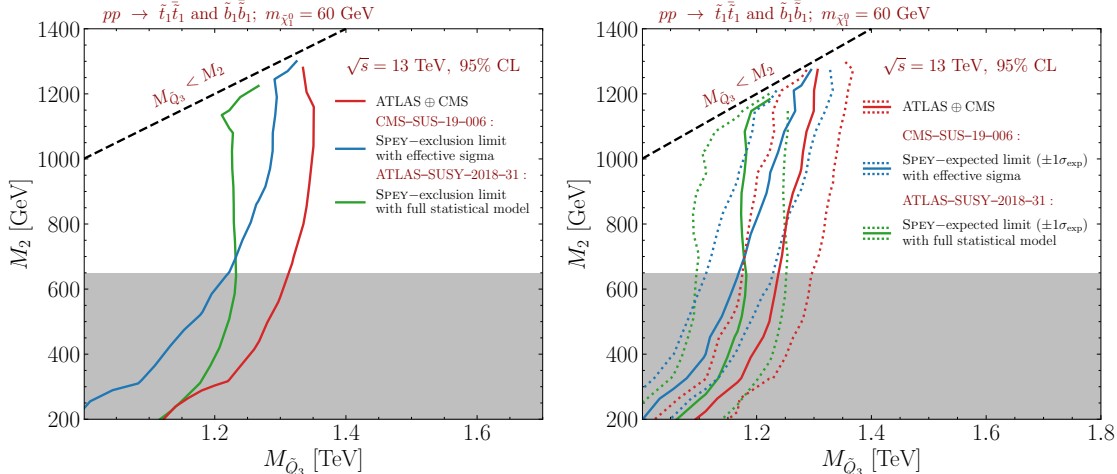

Figure 6: *Left panel shows the observed exclusion limit generated by correlated background model in CMS-SUS-19-006 analysis (blue) and full statistical model in ATLAS-SUSY-2018-31 analysis (green). The red curve represents the new limit that can be achieved by combining these PDF distributions. The right panel shows the same for expected exclusion limits along with one standard deviation, which is captured via dotted lines. The grey area has been excluded by electroweakino searches.*

A closer examination of the profiled likelihood distributions in three different scenarios can be achieved by analyzing the $\chi^2(\mu)$ distribution defined in equation (12). Figure 7 provides a visualization of the $\chi^2(\mu)$ distribution for $\mu \in [0, 2.2]$, utilizing the same color scheme as in Figure 6. The dashed red line represents the $\chi^2(\mu)$ value corresponding to the upper limit of the parameter of interest (POI) in the combined scenario. The exclusion cross-section, indicated by the same colour code, demonstrates that the combined scenario reduces the expected upper limit on the cross-section by 26%. Additionally, the shaded yellow area represents the $\pm 1\sigma$ deviation from the background hypothesis.

## 4 An example beyond LHC: Neutrino mass ordering problem

The capabilities of SPEY have predominantly been demonstrated within the confines of the LHC experiments. However, it is crucial to highlight that any empirical study necessitates hypothesis testing. In order to showcase the versatility of SPEY, we aim to partially reproduce the findings presented in ref. [55], which addresses the issue of neutrino mass ordering.

The neutrino mass ordering problem pertains to the arrangement of neutrino mass eigenstates, specifically whether they follow a normal ordering ($m_{\nu_1} < m_{\nu_2} < m_{\nu_3}$) or an inverted ordering ($m_{\nu_3} < m_{\nu_1} < m_{\nu_2}$). In ref. [55], the authors analyze the latest data from the T2K and NOvA experiments, which measure the oscillation probabilities of muon and electron neutrinos and antineutrinos. Their analysis reveals a significant reduction in the preference for normal ordering compared to previous results, indicating that the mass ordering problem remains unresolved. The study also explores the implications of these findings for other neutrino physics investigations, including the JUNO, DUNE, and T2HK experiments, which are expected to provide a high-confidence determination of the mass ordering problem.

In this study, we reproduce their results based on the T2K experiment, which requires extensive computation based on energy flux, where details can be found in refs. [55, 56]. The expected number of events has been computed by considering different parameters related to

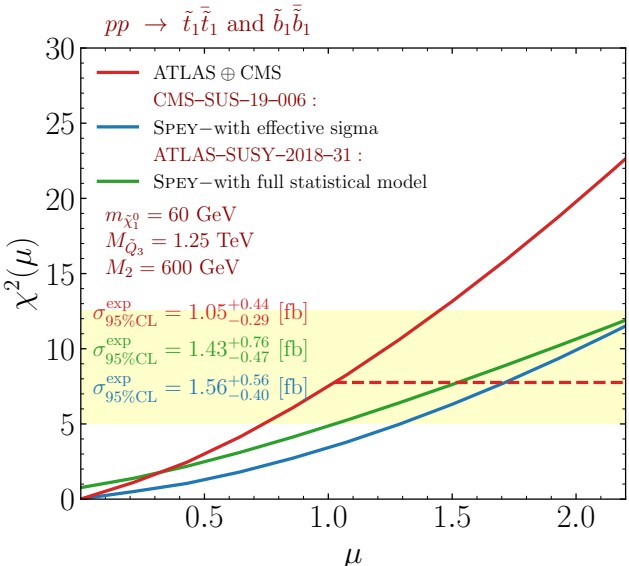

Figure 7: $\chi^2(\mu)$ *versus POI distribution for a benchmark point at $M_2 = 600$ GeV, $M_{\tilde{Q}_3} = 1.25$ TeV and $m_{\tilde{\chi}_1^0} = 60$ GeV. ATLAS-SUSY-2018-31 results are plotted with the full statistical model (green), and CMS-SUS-19-006 results are computed with a correlated background model (blue). The red curve represents combined likelihood. The dashed red line represents the $\chi^2$ value at the 95% CL upper limit of POI, and the shaded area within dotted lines shows $1\sigma$ deviation from the expected background for the combined scenario.*

the degree of the CP violation ($\delta_{CP}$), mixing angle ($\phi_{23}$),[20] and the mass difference between neutrino flavours ($\Delta m_{31}^2$). The range for $\delta_{CP}$ was set to $[-\pi, \pi]$, while $\sin^2 \phi_{23}$ varied between 0.35 and 0.64, and $\Delta m_{31}^2$ fell within the interval $[2.4, 2.7] \times 10^{-3}$ eV$^2$ for normal ordering and $[-2.7, -2.4] \times 10^{-3}$ eV$^2$ for inverse ordering. The observed data and expected background yields for the T2K experiment were obtained from ref. [57].

Based on the information provided, we formed a likelihood backend for SPEY interface following the guidelines presented in Appendix A with the following functional form:

$$\mathcal{L}(\mu, \theta) = \left[ \prod_{i \in \text{channels}} \prod_{j_i \in \text{bins}} \text{Poiss}\left(n^j \middle| (\mu n_s^j + n_b^j)(1 + \theta^j \sigma_b^j)\right) \right] \cdot \prod_{k \in \text{nuis}} \mathcal{N}\left(\theta^k \middle| 0, 1\right), \quad (16)$$

where signal yields is a function of $n_s \equiv n_s(\delta_{CP}, \sin^2 \phi_{23}, \Delta m_{31}^2)$. The likelihood function used in this study is defined as the product of bins within a set of channels. The provided dataset consists of 5 distinct channels, with each channel containing 40 bins. While the original implementation employs $\mathcal{N}\left(\theta^k \middle| 0, \sigma_b\right)$, we introduced a reparameterization of the likelihood to enhance the optimization process and achieve a constraint term that follows a unit-Gaussian distribution.

The results of the $\Delta \chi^2$ fit are illustrated in Fig. 8. $\Delta \chi^2$ is defined as

$$\Delta \chi^2 = -2 \log \frac{\mathcal{L}\left(1, \theta \middle| \delta_{CP}, \sin^2 \phi_{23}, \Delta m_{31}^2\right)}{\mathcal{L}_{\max}\left(1, \theta \middle| \delta_{CP}, \sin^2 \phi_{23}, \Delta m_{31}^2\right)},$$

---

[20]Note that usually the mixing angle is shown with $\theta_{23}$ but in order to not confuse the mixing angle with the nuisance parameters, we will stick with $\phi_{23}$ notation.

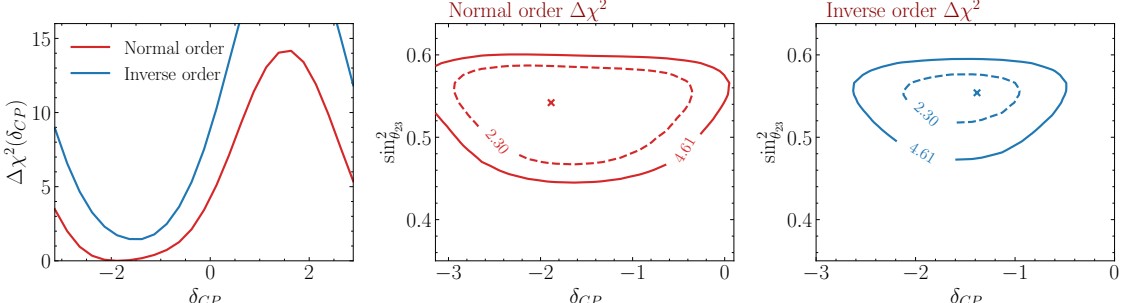

Figure 8: *Left panel shows $\Delta\chi^2$ distribution versus $\delta_{CP}$ for normal mass ordering (red) and inverse mass ordering (blue). The middle panel shows $\Delta\chi^2$ contour for the marginalised likelihood for normal mass ordering in $(\sin^2\phi_{23}, \delta_{CP})$-plane, and the right panel shows the same for inverse mass ordering. The cross in the middle and right panels indicates the best-fit point.*

where $\mathcal{L}\left(\mu, \theta | \delta_{CP}, \sin^2\phi_{23}, \Delta m_{31}^2\right)$ is marginalised over $\sin^2\phi_{23}$ or $\Delta m_{31}^2$; hence POI has been set to one for both terms. In the left panel, the $\Delta\chi^2$ distribution is presented with respect to $\delta_{CP}$, obtained by marginalizing the likelihood with respect to $\sin^2\phi_{23}$ and $\Delta m_{31}^2$ for a fixed $\delta_{CP}$. In order to slightly favour the normal order, the maximum likelihood of the inverse ordering distribution was adjusted to equation with the maximum likelihood of the normal ordering scenario. The middle (right) panel displays the $\Delta\chi^2$ contours at significance levels of 2.3 and 4.61 for the normal ordering (inverse ordering) case. The marginalization was carried out solely over $\Delta m_{31}^2$ by fixing $\delta_{CP}$ and $\sin^2\phi_{23}$. The cross depicted in each plot represents the best-fit location for the respective scenario.

Our findings equation closely with the results reported in ref. [55] (refer to Fig. 3 in the reference), thus validating the performance of SPEY interface across a broad spectrum of applications. The implementation for eq. (16) can be accessed from this GitHub repository.

# 5 Conclusion & future directions

SPEY is a modular, cross-platform Python package for building statistical models for reinterpretation studies. It allows the integration of likelihood prescriptions to be used through a global inference system without the need to alter the structure of the package. This has been achieved through an auto plug-in detection system, which searches through Python packages to find plug-in entry points. Such construction allows for a likelihood prescription agnostic interface which can be extended to cover any future likelihood constructions, which may enable a more accurate approximation of the full likelihood.

Whilst the publication of full statistical models reincarnates the experimental analysis, it can be computationally highly complex for fast inference. To improve computational limitations of full statistical models, machine-learned likelihood prescriptions have emerged [58,59]. Additionally, methodologies such as simplified likelihoods using linearised systematic uncertainties [60] have significantly improved the computational cost of the full statistical models. SPEY's modular construction enables the usage of such likelihood prescriptions, allowing highly efficient full likelihood estimations to be used for reinterpretation studies through one package. Such plug-in integrations are currently in the making.

Combining different statistical models provides a valuable meta-analysis framework despite the challenges. Using overlap removal techniques [20] are highly effective despite their limitations. Application of methodologies used in Higgs physics, such as Simplified Template

Cross Sections [61] (STXS), can allow for more robust likelihood combinations due to non-overlapping phase spaces. SPEY's backend agnostic structure enables the combination of any likelihood prescription, originating from dedicated packages, seamlessly, and we will continue the development of new approaches for combinations.

SPEY is currently being implemented into SMODELS [11] and MADANALYSIS 5 [5–9] packages to be used for inference which will be available in the future releases. In the future, we are planning to include tools like CONTUR [62], Rivet [1] and CheckMATE [10] to achieve a reinterpretation software agnostic platform. This will allow searches and measurements to be used together across multiple platforms to enhance the limits of meta-analysis.

In this study, we have demonstrated the diverse range of use cases and applications of the SPEY package, spanning from LHC to neutrino experiments. Our contributions include the introduction of a novel simplified likelihood scenario, leveraging an effective variance that surpasses the performance of previous simplified likelihood approaches. Moreover, we have developed a user-friendly likelihood combination interface capable of seamlessly integrating full statistical models while ensuring compatibility with their respective properties. Additionally, we have devised a method to effectively combine statistical models lacking sufficient information regarding the source of nuisance parameters.

## Acknowledgments

We express our sincere gratitude to Sabine Kraml and Wolfgang Waltenberger for fruitful discussions and suggestions. We would also like to thank Timothée Pascal for his dedicated efforts in integrating and testing SPEY within the SModelS package. Our thanks go to William Ford for providing valuable insights into their recasting endeavours for the CMS-SUS-20-004 analysis, as well as Yuber F. Perez-Gonzalez for sharing detailed information about his work on reinterpreting neutrino experiments. We are deeply grateful to the pyhf development team, namely Lukas Heinrich, Matthew Feickert, Giordon Stark, and Alexander Held, for their invaluable support in addressing numerous inquiries related to the package. Finally, we would like to express our appreciation to Benjamin Fuks and Andy Buckley for their extensive feedback on the manuscript.

## A  Building a backend plug-in

To create a new PDF prescription for SPEY to use, one needs to import a few tools first. The first step in creating your own SPEY plug-in is to create a statistical model interface. This is as simple as importing abstract base class `BackendBase` from SPEY and inheriting it. Let us implement an example of

$$\mathcal{L}(\mu) = \prod_{i \in \text{bins}} \text{Poiss}\left(n^i | \mu n_s^i + n_b^i\right),$$

which is a simple likelihood prescription described by Poisson distribution without any uncertainties. As before, $n$, $n_s$, and $n_b$ refers to data, signal and background yields, respectively.

```
1  from scipy.stats import Poisson
2  import numpy as np
3  from spey import BackendBase, ExpectationType
4  from spey.base.model_config import ModelConfig
5
6  class PoissonPDF(BackendBase):
7      """Example Poisson plug-in"""
```

```
8
9      name = "example.poisson"
10     version = "1.0.0"
11     author = "Tom Bombadil <tom@bombadil.com>"
12     spey_requires = ">=0.0.1,<0.1.0"
13     arXiv = ["abcd.xyzw"]
14     doi = ["doi/address"]
15
16     def __init__(
17         self, signal_yields, background_yields, data
18     ):
19         self.signal_yields = signal_yields
20         self.background_yields = background_yields
21         self.data = data
```

Before writing the initialisation routine, we included metadata that SPEY needs in order to correctly identify the module, and then we simply initialise it with signal, background and data yields. The first three metadata sections indicates the name of the plugin (`name`), its version (`version`), and author information (`author`). Is followed by `spey_requires` which informs SPEY to execute this particular example only between SPEY versions 0.0.1 and 0.1.0 which protects other users in case of significant changes within SPEY interface which might affect the usage of our new plug-in. Following metadata for `arXiv` and `doi` is to ensure that our new plug-in is cited properly. For details on how SPEY reports on metadata information, see App. B.

In the following, we add two functions to inform SPEY about the structure of the statistical model data

```
1      @property
2      def is_alive(self) -> bool:
3          return np.any(self.signal_yields > 0.0)
4
5      def config(
6          self,
7          allow_negative_signal=True,
8          poi_upper_bound=10.0
9      ):
10         min_poi = -np.min(
11             self.background_yields[self.signal_yields > 0]
12             / self.signal_yields[self.signal_yields > 0]
13         )
14
15         return ModelConfig(
16             0, # poi index location
17             min_poi,
18             [0.0], # initialisation suggestion for the optimiser
19             [(min_poi if allow_negative_signal else 0.0,
20                 poi_upper_bound)], # bound suggestion for the
                    optimiser
20         )
```

The first function serves to provide SPEY with essential information regarding the signal, allowing it to bypass unnecessary computations. On the other hand, the second function involves configuring the statistical model. This configuration entails defining the minimum value the parameter of interest (POI) can assume without resulting in negative expected yields ($\mu_{min} = -\min(n_b/n_s)$). Additionally, it specifies the index of the POI within the parameter list (for simplicity, we assume $\mu$ to be the first parameter), initialization values for both the POI and nuisance parameters, as well as their respective bounds to be used during the minimization of the negative log-likelihood. Incorporating a maximum allowable value for the POI aids the optimization process by avoiding regions where the likelihood is undefined, thus significantly improving execution time. It is worth noting that, while this specific example does not involve

multiple parameters (i.e., nuisance parameters), the location index of the POI becomes crucial when nuisance parameters are present.

In the following, we implement the accessor to the expected data

```
def expected_data(self, pars, **kwargs):
    return pars[0] * self.signal_yields + self.background_yields
```

which returns the mean value of the Poisson distribution for a given POI.

Finally one needs to define the function to compute log-probability distribution ($\log \mathcal{L}(\mu)$) which is defined using `get_logpdf_func` function.

```
def get_logpdf_func(
    self,
    expected = ExpectationType.observed,
    data = None,
):
    current_data = (
        self.background_yields if expected == ExpectationType.
            apriori else self.data
    )
    data = current_data if data is None else data

    return lambda pars: np.sum(
        poisson.logpmf(data, pars[0] * self.signal_yields + self
            .background_yields)
    )
```

Notice that we always modify the data with respect to the `expected` input to make sure that output is computed accordingly. The `data` input is to ensure that computation has been done correctly in case of computing the exclusion limits by sampling through the likelihood or while computing the Asimov likelihoods.

Once the implementation is complete one needs to write `setup.py` including

```
from setuptools import setup
setup(
    entry_points={
        "spey.backend.plugins": [
            "example.poisson = example_poisson:PoissonPDF"
        ]
    }
)
```

where setup function creates an entry point within `"spey.backend.plugins"` collection and it informs SPEY about the location of the implementation which lives in `example_poisson.py` file with the class name `PoissonPDF`. Once the package is installed using `pip install -e .` command in the terminal, SPEY can automatically detect and use it. A simple test then can be done via following

```
import spey
import numpy as np

stat_wrapper = spey.get_backend("example.poisson")
stat_model = stat_wrapper(
    signal_yields= np.array([12,15]),
    background_yields = np.array([50.,48.]),
    data=np.array([36,33])
)
print(stat_model.exclusion_confidence_level())
```

should return `[0.9999807105228611]`. The code for the full implementation can be found in this GitHub repository.

Note that this implementation shows only a fraction of the possible functionalities. A full list of functions can be found in Table 3.

Table 3: List of available functions that can be implemented for SPEY to use the new PDF prescription.

| Functions and Properties | Explanation |
| --- | --- |
| `is_alive` (property) | Is any of the regions provided within the statistical model is non-zero. |
| `config` | Configuration of the statistical model, informing SPEY about the boundaries, initial values and the location of the parameter indices. |
| `expected_data` | (optional) The mean value of the likelihood for given parameters. |
| `get_logpdf_func` | The definition of the log-probability distribution. |
| `get_objective_function` | (optional) If the objective function is different than negative log-likelihood or gradients can be computed for the optimisation process, this function allows SPEY to pass that information to the optimiser. |
| `get_hessian_logpdf_func` | (optional) This function provides the necessary computing tools to SPEY to compute the Hessian of the log-probability distribution. |
| `get_sampler` | (optional) Generates a function to sample from the likelihood distribution. |

## B  Citing backends

SPEY ensures that the proper citation information for each backend has been included within the class metadata. In order to access metadata information for each backend, one can use `spey.get_backend_metadata` function, which for instance will return the following for `"default_pdf.third_moment_expansion"`

```
1  spey.get_backend_metadata("default_pdf.third_moment_expansion")
2  {"name": "default_pdf.third_moment_expansion",
3   "author": "SpeysideHEP",
4   "version": "0.0.1",
5   "spey_requires": "0.0.1",
6   "doi": ["10.1007/JHEP04(2019)064"],
7   "arXiv": ["1809.05548"]}
```

This provides the information from top to bottom, name of the backend, author of the backend, version of the backend, the SPEY version that the backend requires, list of DOI and arXiv numbers.

## C  Third moments from asymmetric uncertainties

Assuming that the uncertainties are modelled as Gaussian (which is a fair assumption since all the uncertainties presented in eq. (3) are Gaussian), third moments of the asymmetric uncertainties can be computed by integrating over Bifurcated Gaussian

$$m^{(3)} = \frac{2}{\sigma^+ + \sigma^-}\left[\sigma^- \int_{-\infty}^0 x^3 \mathcal{N}(x|0,\sigma^-)dx + \sigma^+ \int_0^\infty x^3 \mathcal{N}(x|0,\sigma^+)dx\right], \qquad \text{(C.1)}$$

where $\sigma^\pm$ represents upper and lower uncertainty envelops and $\mathcal{N}(x|0,\sigma)$ is the normal distribution. This computation can be achieved by importing `compute_third_moments` function

```
from spey.backends.simplifiedlikelihood_backend.third_moment import
    compute_third_moments
third_moments = compute_third_moments(upper_envelops,
    lower_envelops)
```

where `upper_envelops` and `lower_envelops` are NumPy arrays with same shape as background. Note that third-moment expansion presented in simplified likelihood framework is only valid if $8\text{diag}(\Sigma)_i^3 \geq (m_i^{(3)})^2$ where $\Sigma$ is the covariance matrix. For the terms that this inequality does not hold SPEY will warn the user and set such terms to zero.

Notice that eq. (C.1) has been derived from the expectation value of n-th moment

$$\mathbb{E}[(X-c)^n] = \int_{-\infty}^\infty (x-c)^n f(x)dx,$$

where $c$ is the shift from the central value and $f(x)$ is the model.

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
