# Peer review of "Spey: smooth inference for reinterpretation studies"

_SciPost Physics, doi:SciPost Phys. 16, 032 (2024)_

## Round 2 · Referee Report · Anonymous (Referee 1) · 2023-10-19

Report

In this paper, the author presents a new software tool designed for statistical inference in reinterpretation studies. The tool's versatility enables users to create various likelihood approximations, integrate pre-existing packages, and combine these likelihoods under multiple assumptions. Additionally, the author introduces a new likelihood approximation technique to improve the precision of previously proposed simplified approaches. All techniques are rigorously tested throughout the paper, and their respective limitations are clearly delineated. The improvements and techniques provided in this paper have been showcased to enhance the limits of an MSSM scenario, demonstrating that even with existing analyses, one can extend the boundaries of current searches.

Furthermore, the author allows users to extend the coverage of the package in terms of the versatility of likelihood proposals and adds the means to cite their independent extensions through the package. This feature enhances the tool's utility and promotes collaboration and the sharing of new statistical methodologies. Hence, I believe the manuscript is valuable for HEP and any empirical sciences.

I have a minor structural observation concerning the paper. While the conclusion is well-structured, providing a concise summary and outlining future steps, a similar organisation is missing in the introduction. I recommend that the author revise the introduction to underscore the current challenges within the field and elucidate how this tool offers a viable solution to these issues.
  • validity: -
  • significance: -
  • originality: -
  • clarity: -
  • formatting: -
  • grammar: -

Author:  Jack Araz  on 2023-12-04  [id 4168]

(in reply to Report 1 on 2023-10-19)

I thank the referee for their analysis of the paper; I have updated the introduction in accordance with the referee's request, which can be found in Arxiv v3. I hope this version will satisfy the referee for the publication of the manuscript.

Thanks
Best regards

---

## Round 2 · Referee Report · Anonymous (Referee 2) · 2023-10-29

Strengths

Presents a tested and accessible solution to a common and important set of problems in the field.

Is well written and clear and comes with links to well-documented open-source code.

Weaknesses

Focusses too much on exclusion limits, at least in the introduction. More emphasis early on on potential discovery, and on fitting parameters (as shown in the later neutrino example) would improve the paper.

Report

Reproducible interpretation and re-interpretation of statistical results based on experimental data is a topic of growing and crucial importance. If sucessfully adopted and supported, this package could facilitate significantly better communication and rigour in this area.

Requested changes

  1. See my comment on the introduction above.
  2. Eq 2 and 3 are a bit confusing in terms of the subscript/superscript on n. The index i moves from subscript to superscript, and seems to maybe sum over s and b at one point and over bins at the next. Also, lambda is said to be a function of n_s and n_b but is shown as a function of mu and theta. I realise this is pretty standard stuff but please check for clarity. 3 page 5 please clarify what "units determined by the user" means. How are these units used and reported by Spey? Does Spey even know or care what they are?

  • validity: high
  • significance: high
  • originality: good
  • clarity: high
  • formatting: excellent
  • grammar: good

Author:  Jack Araz  on 2023-12-04  [id 4167]

(in reply to Report 2 on 2023-10-29)

I thank the referee for the detailed analysis of the paper. I have updated the manuscript in accordance with their comments, which can be found in Arxiv v3.

1) I have updated the introduction and included an additional example in section 2 to emphasize these. 2) All indices have been moved to superscript, and the definition of lambda has been extended. Additionally, I added a footnote stating that Spey does not need this particular definition, i.e. lambda can be a function of anything since this is defined by the user. Still, current implementations are only accepting a number of yields so far. 3) This has been clarified in footnote 8; indeed, Spey does not care/know the unit of the cross-section. This implementation originated from a request from recasting packages to compute the upper limit on cross-section instead of only on POI (which is essentially the upper limit on POI times cross-section value, hence can easily be computed externally using the poi_upper_limit() function).

I hope these corrections and clarifications will satisfy the referee for the publication of the manuscript.

Thanks Best regards

---

## Round 3 · List of Changes

• Abstract has been updated.
  • The introduction has been updated.
  • Eq 2 and the discussion around it has been updated.
  • An example has been added at the end of section 2.

---

## Editorial Decision

published